# Diagnosis of Cervical Cancer and Pre-Cancerous Lesions by Artificial Intelligence: A Systematic Review

**DOI:** 10.3390/diagnostics12112771

**Published:** 2022-11-13

**Authors:** Leila Allahqoli, Antonio Simone Laganà, Afrooz Mazidimoradi, Hamid Salehiniya, Veronika Günther, Vito Chiantera, Shirin Karimi Goghari, Mohammad Matin Ghiasvand, Azam Rahmani, Zohre Momenimovahed, Ibrahim Alkatout

**Affiliations:** 1Midwifery Department, Ministry of Health and Medical Education, Tehran 1467664961, Iran; 2Unit of Gynecologic Oncology, ARNAS “Civico-Di Cristina-Benfratelli”, Department of Health Promotion, Mother and Child Care, Internal Medicine and Medical Specialties (PROMISE), University of Palermo, 90127 Palermo, Italy; 3Neyriz Public Health Clinic, Shiraz University of Medical Sciences, Shiraz 7134814336, Iran; 4Social Determinants of Health Research Center, Birjand University of Medical Sciences, Birjand 9717853577, Iran; 5University Hospitals Schleswig-Holstein, Campus Kiel, Kiel School of Gynaecological Endoscopy, Arnold-Heller-Str. 3, Haus 24, 24105 Kiel, Germany; 6School of Industrial and Systems Engineering, Tarbiat Modares University (TMU), Tehran 1411713114, Iran; 7Department of Computer Engineering, Amirkabir University of Technology (AUT), Tehran 1591634311, Iran; 8Nursing and Midwifery Care Research Centre, School of Nursing and Midwifery, Tehran University of Medical Sciences, Tehran 141973317, Iran; 9Reproductive Health Department, Qom University of Medical Sciences, Qom 3716993456, Iran

**Keywords:** artificial intelligence, prediction, screening, cytology, colposcopy, cervical cancer

## Abstract

Objective: The likelihood of timely treatment for cervical cancer increases with timely detection of abnormal cervical cells. Automated methods of detecting abnormal cervical cells were established because manual identification requires skilled pathologists and is time consuming and prone to error. The purpose of this systematic review is to evaluate the diagnostic performance of artificial intelligence (AI) technologies for the prediction, screening, and diagnosis of cervical cancer and pre-cancerous lesions. Materials and Methods: Comprehensive searches were performed on three databases: Medline, Web of Science Core Collection (Indexes = SCI-EXPANDED, SSCI, A & HCI Timespan) and Scopus to find papers published until July 2022. Articles that applied any AI technique for the prediction, screening, and diagnosis of cervical cancer were included in the review. No time restriction was applied. Articles were searched, screened, incorporated, and analyzed in accordance with the Preferred Reporting Items for Systematic Reviews and Meta-analyses guidelines. Results: The primary search yielded 2538 articles. After screening and evaluation of eligibility, 117 studies were incorporated in the review. AI techniques were found to play a significant role in screening systems for pre-cancerous and cancerous cervical lesions. The accuracy of the algorithms in predicting cervical cancer varied from 70% to 100%. AI techniques make a distinction between cancerous and normal Pap smears with 80–100% accuracy. AI is expected to serve as a practical tool for doctors in making accurate clinical diagnoses. The reported sensitivity and specificity of AI in colposcopy for the detection of CIN2+ were 71.9–98.22% and 51.8–96.2%, respectively. Conclusion: The present review highlights the acceptable performance of AI systems in the prediction, screening, or detection of cervical cancer and pre-cancerous lesions, especially when faced with a paucity of specialized centers or medical resources. In combination with human evaluation, AI could serve as a helpful tool in the interpretation of cervical smears or images.

## 1. Introduction

Cervical cancer is the fourth most frequently diagnosed cancer and the fourth leading cause of cancer death in women [1], which accounts for 570,000 incident cases and 310,000 deaths each year worldwide [2]. Cervical cancer is the most commonly diagnosed cancer in 23 countries in the world and is the leading cause of cancer death in 36 countries, with the vast majority of these countries found in sub-Saharan Africa, Melanesia, South America, and South-Eastern Asia. The region with the highest incidence and mortality rates is sub-Saharan Africa, with rates particularly high in Eastern Africa (Malawi has the world’s highest incidence and mortality rate in the world), Southern Africa, and Middle Africa. In Northern America, Australia/New Zealand, and Western Asia (Saudi Arabia and Iraq), incidence rates are 7 to 10 times lower, while mortality rates can vary by up to 18 times [3].

Cervical cancer is almost fully preventable because of its long premalignant phase and is curable if the signs are identified and treated in the earliest stage [4,5,6]. Effective prevention strategies for cervical cancer include the use of screening tests for primary and secondary prevention, human papillomavirus (HPV) vaccination, and early detection and treatment of cervical intraepithelial neoplasia. [7,8,9,10,11,12,13,14]. The gold standard for the diagnosis of cervical cancer is still colposcopy-guided biopsy, which is then staged on the basis of the clinical examination and the outcome of imaging procedures [15,16]. Although this is an effective strategy in high-income nations, its execution requires a well-organized infrastructure, as well as skilled professionals, such as pathologists, cytopathologists, and colposcopy specialists [17,18,19]. Many efficient screening programs have been developed in recent times [20], but their implementation and maintenance are hindered by insufficient numbers of professionals [21] and the absence of an adequate health infrastructure [22]. Furthermore, manual screening is not always accurate [23], and may cause some lesions to escape detection for a certain period of time [16].

Artificial intelligence (AI) provides an automated diagnosis; its significant potential in resolving this issue is proven [24]. AI has been used to an increasing extent for the diagnosis of numerous diseases in recent years, including the classification of skin malignancies [25], imaging diagnosis of tumors [26], the detection and classification of retinal diseases [27], and gynecologic cancer [28]. With the use of sophisticated algorithms, AI has the ability to recognize images, learn classifications, extract features, and process data autonomously [16,29]. The research suggests that AI-assisted technology may be utilized for segmentation of cytoplasm and identification of cervical epithelial dysplasia; however, it is still unknown how well AI-assisted cytology will perform in population-based screening [30]. The application of AI in early screening and detection of cervical cancer has helped address the issue of scarce human resources and enhances diagnostic precision [16]. While most of the technologies supporting AI in pathology are either in the development stage or are in the observational study stage, they are not frequently used in routine large-scale screenings [31]. Considering the high demands and expectations of these tools in clinical practice, evidence supporting AI-based diagnosis needs to be systematically reviewed. The aim of the present review is to evaluate the diagnostic performance of AI technologies in cervical cancer and pre-cancerous lesions.

## 2. Materials and Methods

Published articles that developed or validated AI techniques for the prediction, screening, and diagnosis of cervical cancer were searched for a systematic review (PROSPERO registration ID: CRD42022352650).

### 2.1. Research Question

The review was based on the following research question: what do we know about AI, and what is the accuracy of AI techniques in detecting cervical cancer or pre-cancerous lesions?

### 2.2. Search Strategy and Information Sources

The Preferred Reporting Items for Systematic Reviews and Meta-Analyses (PRISMA) criteria were used to design the study [32]. We searched for relevant articles in the three databases PubMed/MEDLINE, Scopus, and Web of Science. The following keywords were used for the search in July 2022: “artificial intelligence”, combined with “cervical cancer”, “cervical intraepithelial neoplasia (CIN)”, “uterine cervical neoplasms”, “cervical neoplasm”, “atypical squamous cells of undetermined significance”, “high-grade squamous intraepithelial lesion (HSIL)”, “low-grade squamous intraepithelial lesion (LSIL)”, “atypical squamous cells of undetermined significance”, “Pap smear”, “Bethesda system”, “screening”, “prediction”, “cytology”, “colposcopy”, “detection”, and “diagnosis”. MeSH keywords and Boolean (AND, OR) operators were employed to enhance the selection of entries.

### 2.3. Inclusion and Exclusion Criteria

We included all types of observational studies dealing with cervical cancer screening, diagnosis, and prediction programs for early and appropriate detection, performed anywhere in the world and published exclusively in the English language.

Studies focused on animals, genomic and molecular studies, segmentation of nuclei, genomic profiles, biomarkers, chromosomal alteration progression, gene expression profiling, optoelectronic sensor, spectroscopy, mathematical model, and cervical cancer prognosis were excluded. Furthermore, studies addressing the segmentation of cancerous lesions on magnetic resonance imaging (MRI)/computed tomography (CT) images, non-full-text articles, case reports, reviews, commentaries, letters to editors, conference presentations, and reports were omitted. Two authors independently assessed the titles and/or abstracts of articles that were retrieved using the search method and those from additional sources to find studies that would fit the criteria of this systematic review. Discussion with a third (external) collaborator was used to settle any differences they had on the eligibility of specific articles.

### 2.4. Study Selection

Three authors reviewed, screened, and extracted the search results. The EndNote software (EndNote X9, Thomson Reuters) was used to list the studies and screen them on the basis of the inclusion criteria. The third expert author strategy was used for conflicts in the screening step. The studies were initially chosen according to the relevance of their titles and abstracts (LA and AMM). Then, the full texts were examined to validate their eligibility (LA and IA). We included articles that addressed any diagnosis of cervical cancer and pre-cancerous lesions.

### 2.5. Data Extraction and Synthesis

Details of all articles were extracted and reported from the studies using a pre-piloted customized standard form in order to ensure the consistency of this step for all investigations. Data such as the authors, year of publication, sample size, methods, and datasets were independently extracted by the two writers. Any disagreement was clarified through discussion (with a third external collaborator if necessary). Due to the diverse modes of reporting, we performed a narrative synthesis of the studies.

## 3. Results

### 3.1. Search Results

A total of 2538 publications, 327 of which were duplicate articles, were found in the various databases. After reviewing the titles and abstracts of the remaining papers, 1100 were excluded. Of the remaining articles, 994 were omitted for the lack of alignment with the objectives of the study. Finally, the systematic review comprised 117 studies (Figure 1).

#### 3.1.1. Application of AI for Cervical Cancer and Its Cost-Effectiveness

From 1997—when cervical cancer screening was performed for the first time with AI—until today, various machine learning algorithms have been applied for the detection of cervical cancer [30,33,34,35,36,37,38,39,40,41,42,43,44,45,46,47,48,49,50,51,52,53,54]. Common machine learning (ML) models included deep learning (DL), k-nearest neighbors (KNN), artificial neural network (ANN), decision tree (DT), random forest (RF), support vector machine (SVM), logistic regression (LR), synthetic minority oversampling technique (SMOTE), convolutional neural network (CNN), multilayer perceptron (MLP), deep neural networks (DNN), the PAPNET test, and ResNet (residual neural network or a combination of techniques) [36,39,45,46,55,56,57,58,59,60,61,62,63,64,65,66,67,68,69,70,71,72,73,74,75]. The time taken for training and for the prediction of cervical cancer by each algorithm varied markedly. In Kruczkowski et al.’s study, the time needed to train for the Naïve Bayes and CNN algorithms varied from 7.54 ms to 5320 ms. The prediction time for cervical cancer by the Naïve Bayes and RF algorithms varied from 1.81 ms to 15.5 ms, and the accuracies differed [76]. Elakkiya et al. reported an average of 0.2 s to classify the cervical lesion using a hybrid deep learning technique that combined small-object detection generative adversarial networks (SOD-GAN) and the fine-tuned stacked autoencoder (F-SAE). [77]. In recent years, the rapid progress of AI technologies and the use of combined features have significantly reduced costs, time, the cost of training, and the inference time [52,78,79]. These advancements have also improved the patient’s access to professional pathologists and the prompt delivery of cytology results [78].

However, the cost effectiveness of AI was questioned in some studies because the cost of AI techniques far exceeded that of manual screening. Additionally, commonly used techniques require a considerable database for training the application of models and this constitutes a barrier in the diagnosis of cervical cancer [80].

The following four groups were created from the 117 studies that were deemed suitable for the investigation: AI application in cervical cancer prediction (*n* = 22), AI application in cervical cancer screening (*n* = 25), AI application in cytology (*n* = 44), and colposcopy (*n* = 26) for the detection of cervical cancer.

#### 3.1.2. Application of AI in Predicting Cervical Cancer

Identifying predictors is essential for making precise and meaningful predictions [81]. Multivariate adjustment and multiple-regression techniques were introduced for prediction (that is, for estimating the predicted value of a certain outcome as a function of given values of independent variables) [82]. AI studies using machine learning principles have focused on algorithms to predict cervical cancer [55,56,57,58,59,60,61,62,63,83,84,85,86,87,88,89,90,91,92,93,94,95]. The most important predictors of cervical cancer were age, age at first sexual intercourse, number of sexual partners, pregnancies, smoking, period of smoking (years), hormonal contraceptives, period of use of hormonal contraceptives (years), IUD, period of use of IUD (years), STDs, period of STDs (years), Schiller, Hinselmann, cytology, the presence of 15 high-risk HPV genotypes [55,56,57,58,60,84], social status, marital status, personal health level, education level, and the number of caesarean deliveries [63]. However, according to Mudawi et al., certain characteristics of the patient samples, including the quantity of alcohol consumed and the presence of HIV and HSV2, could not be regarded as reliable predictors [95]. Based on the data shown in Table 1, the accuracy of the algorithms in predicting cervical cancer varied from 70% to 100% [55,56,57,58,59,60,62,83,84,85,86,87,88,89,90,91,92,93,94,95]. The application of AI for the prediction of cervical cancer is shown in Table 1.

#### 3.1.3. Application of AI in Cervical Cancer Screening

Screening is a way of identifying apparently healthy people who may have an increased risk of a particular condition [97]. The screening test needs to be sensitive and precise. A screening test must have sensitivity exceeding 95% if the specificity is less than or equal to 95% and vice versa (specificity must be >95% if the sensitivity is 95%) in order to detect more true-positive cases than false-positive cases when the prevalence of the disease is less than or equal to 5% (which covers the majority of screening populations). Most screening tests do not meet this high standard, which means that the screening program must absorb the costs of many false-positive results [98].

The use of AI in screening for cervical cancer has produced contradictory results. In some studies, the use of artificial intelligence reduced false-negative outcomes compared to traditional methods [33]. Other studies reported an increase in false-negative outcomes [35]. Yet other investigations registered no difference between AI and conventional methods [37]. Michelow and coworkers [34] reported no significant difference between manual detection and PAPNET for invasive carcinoma and HSIL. Interestingly, these contradictions were observed in older studies as well [33,35,37]. In recent times, the performance of AI in cervical cancer screening was better than that of conventional and manual methods [38,42,53]. With equal sensitivity and much higher specificity compared to both Pap and manual DS, AI-based dual staining (DS) yielded lower positivity than cytology and manual DS [53]. The better performance of AI in recent studies may be attributed to the hybrid ensemble approach, combined algorithms and techniques, and the grade of squamous intraepithelial lesions [38,39,46]. Sarwar et al. [38] reported that the hybrid ensemble technique outperformed all other algorithms and demonstrated a screening efficiency of nearly 98%. According to Bao et al. [30], the overall agreement rate between manual reading and AI in CIN detection was 94.7% (95% confidence interval 94.5–94.8%), and the kappa coefficient was 0.92 (0.91–0.92). Furthermore, the performance of AI in the detection of CIN2+ increased with the severity of detected abnormalities on cytology. The accuracy of AI in screening for CIN 1–3 and adenocarcinoma in situ varies between 67% and 98.27% [39,40,41]. The application of AI in cervical cancer screening is shown in Table 2.

#### 3.1.4. Application of AI in Cytology for the Detection of Cervical Cancer

Cervical cytology image analysis is a very time-consuming, challenging, and laborious task [99]. Computer-assisted diagnosis is believed to ease this situation because it can potentially lower the misdiagnosis rate and also reduce the workload of cytologists [100]. Therefore, several studies have addressed the subject of automatic cervical cancer diagnosis [64,65,66,67,68,74,75,80,101,102,103,104,105,106,107,108,109,110,111,112,113,114,115,116,117,118,119,120,121,122,123,124,125,126,127,128,129,130,131,132,133,134,135,136]. The investigations showed that AI-assisted methods were promising, and achieved a high sensitivity and specificity in clinical cervical cytological screening [66,126]. In a multicenter, clinical-based observational study by Bao et al., AI-assisted reading identified considerably more CIN 2 (92.6%) and CIN 3+ (96.1%) lesions than, or at a similar rate as, manual reading. Compared to expert cytologists, AI-assisted reading showed a similar sensitivity (relative sensitivity of 1.01) and greater specificity (relative specificity of 1.26) [66]. Cao and co-workers [130] compared the performance of AI-assisted reading with that of four pathologists. The first two pathologists had 4 years of work experience, and the third and fourth pathologist had 7 and 10 years of work experience, respectively. The proposed model achieved an area under the receiver operating characteristics (AUROC) of 0.99, and an accuracy of 98.0%, which was comparable to a pathologist with a decade of expertise (accuracy, 93.7%). Additionally, pathologists needed on average 14.83 s to diagnose each image, compared to the 0.04 s needed by the AI-assisted method. In fact, reading with AI assistance was approximately 380 times faster than reading by a typical pathologist. AI algorithms were able to distinguish between normal and cancerous Pap smears with an accuracy of 80–100% [68,110,111,113,115,116,119,120,125,127,130,131,135,137]. The application of AI in cytology for the detection of cervical cancer is shown in Table 3.

#### 3.1.5. Application of AI in Colposcopy for the Detection of Cervical Cancer

AI-assisted tools appear to be very suitable for the cervical cancer diagnostic protocol, which recommends colposcopy in cases of an abnormal PAP smear and/or high-risk HPV and the collection of diagnostic tissue samples before initiating any potentially invasive treatment [138]. In response to this demand, a few notable studies were published on the use of AI in colposcopy for the detection of cervical cancer [69,70,71,72,73,77,79,137,138,139,140,141,142,143,144,145,146,147,148,149,150,151,152,153,154,155,156]. According to several investigations, the AI diagnostic approach could support or even potentially replace conventional colposcopy, permit more objective tissue specimen sampling, and reduce the number of cervical cancer cases in developing nations by offering an economical screening option in low-resource settings [137,141]. Some research suggests that AI could help less skilled clinicians to decide whether to perform a cervical biopsy [70,144,145,156]. In addition, AI helped gynecologists to accurately establish the presence of invasive cancer on cervical pathological images diagnosed by AI [156]. According to a large study (over 19,000 patients) performed in China by Xue et al., the agreement between pathology findings and colposcopic impressions graded by the Colposcopic Artificial Intelligence Auxiliary Diagnostic System (CAIADS) was higher than that of colposcopies interpreted by colposcopists (82.2% vs. 65.9%). Additionally, the CAIADS proved to be more accurate in predicting biopsy sites [147]. In published studies, the sensitivity and specificity of AI in colposcopy for the detection of CIN 2 or more severe lesions were reported at 71.9–98.22% and 51.8–96.2%, respectively [70,72,73,137,138,139,140,143,147,148,150,151,152,154]. The accuracy of AI in colposcopy for CIN 2+ detection varied from 40.5% to 98.3% [70,72,77,137,143,146,147,148,150,151,152,154,156]. The use of AI in colposcopy for the detection of cervical cancer is summarized in Table 4.

## 4. Discussion

Despite tremendous progress in the treatment of cancer, cervical cancer cells are occasionally detected at a time when the disease has already caused distant metastasis [157]. This is a major problem in third-world countries with inferior health systems [5,8]. Furthermore, monetary and personal limitations increase the need for alternative tools as the number of people to be treated by professionals increases [158]. Thus, we need better diagnostic tools for cancer [157]. The application of AI not only in medicine but also other majors has grown significantly over the past ten years, particularly in the last five years [28,159,160] at three levels: for patients, by enabling them to process their own data to promote health; for health systems, by improving workflow and the potential for reducing medical errors; and for clinicians, predominantly via rapid, accurate image interpretation [161].

The aim of this systematic review is to evaluate the diagnostic performance of AI technologies in cervical cancer and pre-cancerous lesions. Estimating the prognosis of cervical cancer is one of the most difficult tasks because its management requires a variety of cancer treatment approaches [162,163], which may even impair quality of life to a significant extent [164,165].

The studies included in this systematic review employed models to identify a variety of predictors, including age, numbers of sexual partners, age at the first sexual intercourse, deliveries, smoking, hormonal and barrier contraceptives, STDs, marital status, personal health level, education level, social status, number of caesarean deliveries, and the presence of 15 high-risk HPV genotypes. The accuracy of different AI algorithms in predicting cervical cancer varies from 70% to 100%. More reliable predictions are achieved when the prediction models for cervical cancer are combined with the hybrid ensemble approach. Compared to studies focused on AI techniques, in a cohort study, Schulte-Frohlinde et al. [166] noted that cervical cancer could be predicted among high-risk HPV-positive women. Age at sexual debut was a significant modifier of the incidence of cervical cancer.

The prediction of cervical cancer on the basis of AI techniques has produced promising results. The findings of a study by Nsugbe showed how prediction machines can contribute towards early detection and prioritize the care of patients with cervical cancer, while also allowing for cost-saving benefits when compared with routine cervical cancer screening [167].

Cytology-based cervical cancer screening has poor accuracy [168]. In addition to the fact that the procedure needs clinical consultants who have undergone significant training, it is time consuming and susceptible to human interpretation and error [81,169]. The use of computer technologies may reduce the likelihood of misdiagnoses, analysis time, and assist in early diagnosis [169].

In the present systematic review, we reviewed studies investigating the performance of AI in screening and cytology for the detection of cervical cancer. The accuracy of AI in the detection of CIN 1–3 and adenocarcinoma in situ varied from 67% to 98.27%. However, we registered contradictory results regarding the performance of AI in cervical cancer screening, in terms of poor as well as better performance compared to traditional methods. The poor performance of AI appears to be limited to old methods of AI. Conversely, applying the hybrid ensemble approach and combined applied algorithms of new AI techniques performed better than traditional and manual approaches. We observed that AI techniques are able to distinguish between normal and cancerous Pap smears with 80–100% accuracy, and were 380 times faster than the typical pathologist. Pap smear testing is a fundamental procedure in protecting women from cervical cancer. However, the effort of a cytologist to detect morphologic changes in lesions with 20,000–50,000 cells on a single slide is tedious, arduous, and dependent on experience [67]. In a cross-sectional study by Wergeland Sørbye et al. [168], four pathologists at three hospitals in Norway evaluated one hundred Pap smears (20 cases normal, 20 cases LSIL, 20 cases HSIL, 20 cases atypical squamous cells of undetermined significance (ASC-US), and 20 high grade squamous intra-epithelial lesion (ASC-H)). The accuracy for CIN2+ varied from 74.1% to 83.8%. Therefore, the claim that AI improves the effectiveness of diagnosis, reduces the clinician’s workload, and even enhances the impact of treatment and prognosis would seem plausible [53,138]. Furthermore, AI was shown to be more adept than the human brain in recognizing specific patterns [138].

Currently, the two most common techniques used to diagnose precancerous cervical lesions are colposcopy and guided biopsy. However, numerous investigations have shown that even practitioners who are skilled in colposcopy struggle to make the right diagnosis [170]. Consequently, the standardized and less volatile diagnostic tools of AI might be useful [79]. Many studies included in the present review concluded that, in cervical cancer diagnosis, AI may be able to supplement or perhaps even replace current colposcopy procedures. The sensitivity and specificity of AI for the detection of CIN2+ were reported to be 71.9–98.22% and 51.8–96.2% respectively [70,72,73,137,138,139,140,143,147,148,150,151,152,154]. The accuracy of AI for the detection of CIN 2+ varied from 40.5% to 98.3%. These data demonstrate the potential of AI in reading colposcopic images. According to a meta-analysis by Mitchell et al. [171], the sensitivity of colposcopists in diagnosing CIN varies greatly compared to the performance of AI: the average weighted sensitivity of colposcopy in differentiating between normal and all cervix abnormalities (atypia, low-grade SIL, high-grade SIL, cancer) was 96%, and the average weighted specificity was 48%.

The threshold normal cervix has an area under the ROC curve of 0.80 when compared to other abnormalities. A gynecologist with little experience might overlook high-grade lesions [172]. AI technologies may serve as a practical aid for the inexperienced gynecologist or general physician in making a precise clinic diagnosis or a wise choice in terms of diagnostic intervention, such as whether to perform a punch biopsy or transfer the patient to a specialized center [79,144,171]. According to Kim et al. [79], the clinical interpretation of colposcopic images by AI had a higher AUC in identifying low- and high-risk lesions than the clinical interpretation of colposcopic images by humans. These findings imply that AI interpretations may be used in the clinical setting. A recent study that evaluated deep learning algorithms for automatic categorization of colposcopic images supports this notion [144]. Automated visual evaluation of cervical images had a higher AUC than the original interpretation of cervical images by human or conventional cytology [40].

### Limitations and Recommendations

The limitations of the studies investigated for this systematic review are worthy of mention. First of all, the majority of the investigations were underpowered in regard of the primary outcome because of their small sample sizes. Some algorithms used in the studies are very unstable, which means that a slight change in the data will significantly change the layout of the best decision. Furthermore, some algorithms are slow and needed more memory to run, In fact, millions of observations may be needed for AI techniques to perform acceptably [173]. Second, AI-based models are not widely used in experimental and clinical settings on real datasets. The experimental results on some (small, intermediate and big) machine learning datasets can show the efficiency of the proposed methods, in terms of space, speed, and accuracy [174].

Experimental tests or prospective clinical trials are urgently needed to better highlight the differences between the investigated studies and validate the findings discussed in the present study. Considering the use of different techniques and algorithms in the published studies, it would be meaningful to design a review comparing each technique with others in order to obtain an accurate estimate of the effectiveness of the techniques and establish the potential superiority of the respective methods. According to several studies, the cost-effectiveness of the automated systems is limited, because they are not suitable for use in poorly and moderately developed nations [175]. Some researchers are still working to improve the use of artificial intelligence in cervical cytology. Since a reliable prediction of the clinical outcome would serve as a guide for treatment and the prediction of cervical cancer is most challenging [176], it would be appropriate to specifically address the role of AI in predicting cervical cancer outcomes.

## 5. Conclusions

Our systematic review highlights the acceptable performance of AI systems in the prediction, screening or detection of cervical cancer and pre-cancerous lesions. AI could aid clinicians in making decisions, reducing their workload as well as the likelihood of misdiagnoses. Indeed, AI interpretation of cervical smears or images could serve as an aid when combined with human evaluation. Further studies on prediction and detection are needed for making appropriate decisions about the treatment of cervical cancer. Eventually, this will help to devise programs for the eradication of cervical cancer on a worldwide basis. However, further work will be needed to make AI feasible, reliable, and less expensive for clinical use. The development of novel techniques and algorithms to reduce the impact of data scarcity in the evaluation and prediction of clinical outcomes, as well as the independent validation of machine learning algorithms, may be included in future studies.

## Figures and Tables

**Figure 1 diagnostics-12-02771-f001:**
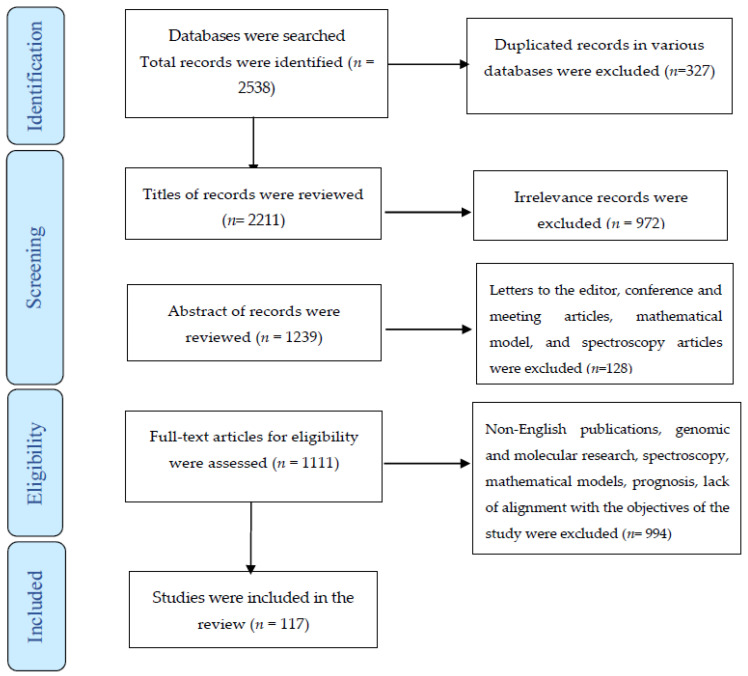
The process of screening and selecting relevant studies based on the Preferred Reporting Items for Systematic Reviews and Meta-Analyses.

**Table 1 diagnostics-12-02771-t001:** Characteristics of studies with AI for the prediction of cervical cancer.

First Author/Year	Sample Size	Methods	Datasets	Main Results	Drawbacks of Studies
Kahng et al., 2015 [55]	731	SVM	Patient records (PAP smear report, age, and the presence of high-risk HPV genotypes)	Four features (PAP, HPV16, HPV52, and HPV35) were found to be the most effective in predicting cancer.	Not reported
Al-Wesabi et al., 2018 [56]	858	DT and KNN	858 samples and 32 features, as well as four classes	When factors including age, first sexual intercourse, pregnancies, smoking, hormonal contraceptives, and genital herpes were taken into account, the accuracy of cancer prediction was 97%.	Not reported
Dillak et al., 2018 [83]	400	RPNN and COA	Subjects (of which 250 were used for training and 150 were used for testing)	The accuracy of the suggested method was 96%.	Not reported
Ahmed et al., 2019 [57]	858	RFE and RF	Patient record (age, age at first sexual intercourse, number of sexual partners, pregnancies, Schiller, Hinselmann, cytology) smoking, smoking in years, IUD, IUD use in years, STDs, years of STDs, and hormonal contraceptives (years).	With an accuracy of 91.04%, this model successfully identified six risk variables for cervical cancer: Schiller, Hinselmann, cytology, first sexual experience (age), number of pregnancies, and age.	Not reported
Alam et al., 2019 [58]	858 patients	DMT and SMOTE	Patient records (age, pregnancies, smoking patterns, chronological records of STDs, and contraceptive usage)	A very high prediction was noted for Boosted DT, which had an AUROC of 0.978.	Not reported
Chen et al., 2019 [59]	365 patients	Boruta algorithm and RF	The age of the patient, uterine cervix images, ThinPrep Pap test, and HPV test	The proposed multi-modal diagnostic approach provides the final diagnosis with 83.1% accuracy	The main limitations of this study were the small sample size and the unbalanced distribution of the patient population
Garg et al., 2019 [84]	-	REPTree	32 essential clinical characteristics, including age, the use of hormonal contraceptives, the number of sexual partners, pregnancies, smoking, etc., as well as four classifications (Hinselmann, Schiller, cytology, and biopsy)	Age, the use of hormonal contraceptives, age at the first sexual encounter, genital herpes. STDs, number of pregnancies, and smoking are the main predictive factors that improve classification in comparison with other factors.	Not reported
Geetha et al., 2019 [60]	858 cases	RF, SMOTE and PCA	Patient data with 32 risk factors and four objective variables: Hinselmann, Schiller, cytology, and biopsy	When factors including age, first sexual encounter, pregnancy, smoking, hormonal contraceptives, and STDs such as genital herpes were taken into account, the accuracy of cancer prediction was 97%.	SMOTE was only applied to two-dimensional data. SMOTE loses effectiveness as dimensions increase since adjacent nodes are not taken into account, leading to overlapping and inaccurate results.
Kar et al., 2019 [85]	15 samples	NFS	Patients’ records	The application of NFS for early-stage detection of cervical cancer produced satisfactory results with 100% accuracy.	Not reported
Kumar Suman and Hooda, 2019 [96]	858 patients	RF, Neural Network, SVM, AdaBoost, Bayes Net, DT	Patient demographics, habits and medical records	The accuracy and AUC of the Bayes Net algorithm were 96.38% and 0.95, respectively.	Not reported
Nithya et al., 2019 [61]	858 patients	C5.0, RF, Rpart, KNN and SVM	Patient data with 36 attributes (32 input features and 4 target variables: Hinselmann, Schiller, cytology, biopsy)	Overall, C5.0 and RF classifiers identified women presenting clinical signs of cervical cancer fairly accurately and thoroughly.	Not reported
Tian et al., 2019 [86]	34 paired samples	MLA (RF)	Adjacent cervical tissues of 14 CIN2+, 10 HPV+ and 10 CIN1 patients.	The probability of accuracy was 0.814 for CIN2+, and 0.922 for HPV+ and CIN1.	The sample size was small.
Alsmariy et al., 2020 [62]	858 cases	SMOTE	32 risk factors (demographic, habits, and historical medical records) with four target variables (Hinselmann, Schiller, cytology, and biopsy)	The accuracy, sensitivity, and PPA ratios of all target variables were increased in the SMOTE voting model by 0.93%, 5.13%, 39.26%, and 29%, respectively.Using the PCA technique shortened the time taken to execute computations and also improved the effectiveness of the model.	Not reported
Asadi et al., 2020 [63]	145 patients	SVM, QUEST, C&R tree, MLP and RBF	Patient data with 23 attributes	The percentages of MLP’s accuracy, sensitivity, specificity, and AUC are 90.90, 90.00, 91.67, and 91.50. The level of personal health, marital status, socioeconomic standing, dose of contraceptives used, education level, and the number of caesarean deliveries were all found to be significant predictors in all algorithms.	Not reported
Ijaz et al., 2020 [88]	858 patients	DBSCAN, SMOTET, RF, iForest	Sexual partners, first sexual encounter, pregnancies, smoking, hormonal contraception, IUDs, STDs, CIN, HPV, and four objective variables: Hinselmann, Schiller, cytology, and biopsy.	DBSCAN with SMOTE and DBSCAN with SMOTETomek were outperformed by combinations of iForest with SMOTE and iForest with SMOTETomek.	Algorithm (which was a combination of outlier technique and became balancing with RF) ran more slowly and required more memory.
Weegar, 2020 [89]	1321 patients with cervical cancer	LSTM neural network	Clinical codes, lab findings, and free text notes on patients, taken from electronic health records.	FR achieved the best results with an AUC of 0.70.	Not reported
Asaduzzaman et al., 2021 [90]	161 patients	ML models	Risk factors for cervical cancer included children, age at first sexual encounter, husband’s age, Pap tests, and age.	The best scores were noted for LR (84.8%) and Sklearn (79.3%).	Not reported
Ilyas et al., 2021 [91]	858 subjects	DT, SVM, RF, KNN, NB, MP, J48 Trees, and LR	Three target variables and cervical cancer risk factors: Hinselmann, Schiller, and cytology	The study shows a high prediction accuracy to 94%, which is significantly higher than the prediction accuracies of individual classification methods tested on the same benchmarked datasets.	Not reported
Jahan et al., 2021 [92]	858 patients’ cases for 32 features	MLP, RF, KNN, DT, LR, SVC, GB, and AdaBoost	Demographics, behaviors, and medical records, as well as four target variables: Hinselmann, Schiller, cytology, and biopsy	Classification models claim the highest accuracy for specific top features such as multilayer perceptrons. The highest accuracy was 98.10% for 30 features.	Not reported
Khan et al., 2021 [93]	858 records	XGBoost, AdaBoost, and RF	Data on 32 risk variables for cervical cancer, including age, cancer, CIN, HPV, and characteristics with no missing values and four targets (Hinselmann, Schiller, cytology, and biopsy)	When compared to 30 features, the performance of the Hinselmann test with the chosen feature produced better results and can be used to diagnose cervical cancer. The accuracy, sensitivity, specificity, PPA, and NPA values for the +e model were 98.83, 97.5, 99.2, 99.17, and 97.63, respectively.	The dataset suffers from huge imbalance, and augmented data was generated using SMOTE.
Mehmood et al., 2021 [94]	858 instances	RF and *shallow neural network*	Demographic data, patient behaviors, and medical history	CervDetect predicted cervical cancer with an accuracy of 93.6%, false-positive and negative rates of 6.4% and 100%, respectively.	Not reported
Mudawi et al., 2022 [95]	585 persons	MLA	Demographics, medical background, and risk factors such as age, IUD use, smoking, STDs, and others.	The RF, DT, adaptive boosting, and gradient boosting algorithms yielded the maximum classification score of 100% for the prediction of cervical cancer. SVM, on the other hand, achieved an accuracy of 99%.	Since the DT method is extremely unstable, even a small change in the data will significantly change the layout of the best decision tree. It is insufficiently reliable SMOTE.

Abbreviations: AUC: area under the ROC curve, CCPM: cervical cancer prediction model, CIN: cervical intraepithelial neoplasia, C&R tree: classification and regression, COA: chaos optimization algorithm, DL: deep learning, DMT: data mining techniques, DT: decision tree, XGBoost: gradient boosting, HPV: human papillomavirus, IUDs: intrauterine devices, KNN: k-nearest neighbors, LSTM: long short-term memory, LR: logistic regression, ML: machine learning, MLA: machine learning algorithm, MLP: multilayer perceptron, NFS: neuro-fuzzy system, NB: naive Bayes, PAP smear: Papanicolaou test, PCA: principle component analysis, QUEST: quick unbiased efficient statistical tree, RF: random forest, RFE: recursive feature elimination, RPNN: ridge polynomial neural network, SMOTE: synthetic minority oversampling technique, STDs: sexually transmitted diseases, SVM: support vector machine.

**Table 2 diagnostics-12-02771-t002:** Characteristics of studies on the use of AI in screening for cervical cancer.

First Author/Year	Sample Size	Methods	Datasets	Main Results
Jenny et al., 1997 [33]	516	PAPNET scan	Women’s cervical smears with abnormal histopathological diagnoses	In conventional screening, the false negative rate fell from 5.7% to 0.8%.
Mango et al., 1998 [35]	Over 10,000	PAPNET vs. conventional microscopic rescreening	Cervical smear	The false negative yield was 6.2% (142/2293) when applying NNA analysis, as opposed to 0.6% (82/13761) when using conventional rescreening.
Michelow et al., 1997 [34]	3106	PAPNET system vs. manual screening	Consecutive normal and abnormal cervical smears	In low-grade lesions, the PAPNET significantly outperformed traditional screening (89.6% vs. 63.8%, respectively). There was no significant difference between PAPNET and manual detection for more serious abnormalities, such as HSIL or invasive cancer (87.5% vs. 94.6%).
Sherman et al., 1998 [36]	7323	PAPNET system vs. conventional microscopic screening	ThinPrep slides of women participating in a population-based study	In the hypothetical scenario, 4.3% and 6.5% of women would have been referred for colposcopy by PAPNET-assisted and manual screening, respectively. Smears taken from women with high-grade SIL or carcinoma were correctly identified by PAPNET-assisted cytological screening.
Nieminen et al., 2003 [37]	108,686	PAPNET system vs. conventional method	Cervical smears	Papnet was able to recognize 92.5% of normal cytologies, while conventional smears had a specificity of 92.9%.
Sarwar et al., 2016 [38]	8091	Novel hybrid ensemble technique	Cervical smears	Algorithms developed using a digital database demonstrated efficiencies in the range of 93% to 95%, whereas multi-class problem algorithms showed efficiencies in the range of 69% to 78%. The hybrid ensemble approach outperformed all other algorithms and achieved an efficiency of approximately 98% for 2-class problems and approximately 86% for 7-class problems.
Kudva et al., 2018 [39]	102	SVM and DT	Digitized cervical images from screening	This algorithm had a sensitivity of 99.05%, specificity of 97.16%, and accuracy of 97.94%.
Hu et al., 2019 [40]	9406	DL-based visual evaluation algorithm	Digitized cervical images from screening	AI identified cumulative precancerous/cancer cases with greater accuracy than conventional cytology ((AUC ¼ 0.91) vs. (AUC ¼ 0.71)).
Sompawong et al., 2019 [41]	1024	Mask Regional CNN (Mask R-CNN)	Pap smear histological slides	The obtained results had a sensitivity, specificity, and accuracy of 72.5%, 94.3%, and 89.8%, respectively.
Bao et al., 2020 [30]	98,549	AI-assisted cytology system vs. manual reading	Pap smear histological slides	Overall, 94.7% of manual readings and AI results concurred. The CIN2+ detection rate increased with the severity of cytological abnormalities, based on both manual reading and AI. AI-assisted cytology was 5.8% more sensitive for CIN2+ detection than manual reading and had a slightly lower specificity than the latter.
Hussain et al., 2020 [87]	1670 images	DL	A hospital-based dataset of Pap smear samples	The suggested method is assessed using three datasets: the Herlev, conventional, and liquid-based cytology datasets. The ensemble classifier produced the best results with 0.989 accuracy, 0.978 sensitivity, and 0.979 specificity.
Hu et al., 2020 [42]	7334	AVE	Cervigram images	By refactoring to a new deep learning-based detection framework, the core AVE algorithm can be operated in approximately 30 s with equivalent accuracy on a basic smartphone. On a low-end smartphone, an image quality algorithm can identify the cervix and evaluate image quality in about one second with an AUC of 0.95 on the ROC curve.
Sahoo et al., 2020 [43]	256	2D MFDFA	Low-coherence images	The specificities and sensitivities between normal and CIN1, CIN1 and CINII, and normal and CIN2 were found to be 94%, 88%, and 93%; and 96%, 98%, and 100% respectively.
Saini et al., 2020 [44]	800	ColpoNet	Colposcopy images	ColpoNet achieved an accuracy of 81.353%. ColpoNet outperformed AlexNet, VGG16, ResNet50, LeNet, and GoogleNet.
Sanyal et al., 2020 [45]	1838	CNN	Microphotographs from cervical smears	The accuracy, sensitivity, specificity, PPV, and NPV by CNN were 95.46%, 94.28%, 96.01%, 91.66%, and 97.30%, respectively. False positives were reported when the CNN failed to recognize overlapping cells (2.7% microphotographs).
Win et al., 2020 [46]	917 Herlev datasets and 966 SIPaKMeD	RF, LD, SVM, KNN, boosted trees, and bagged trees	Pap smear images	Using the SIPaKMeD dataset, the two-class classification accuracy was 98.27%, while the five-class classification accuracy was 94.09%.
Xiang et al., 2020 [47]	1014	YOLOv3	Annotated cervical cell images	On cervical cell image-level screening, the model yielded a sensitivity of 97.5% and a specificity of 67.8%. Produced a cervical cell-level diagnosis with a best mean average precision of 63.4%.
Xue et al., 2020 [48]	3221 women	AVE	7587 filtered images fromMobileODT	For all ROC curves, the AUC values for discrimination of the most likely precancerous cases from the least likely cases were above 0.90. AVE is able to classify images of the cervix with confidence scores that are strongly related to expert evaluations of severity for the same images.
Cheng et al., 2021 [49]	1170 patient-wise	WSI	Cervical smear slides	Achieved 95.1% sensitivity and 93.5% specificity for classifying slides, which compares favorably with the average performance of three independent cytopathologists. Additionally, it was able to identify the top 10 lesion cells on 447 positive slides with an 88.5% true positive rate.
Holmstrom et al., 2021 [31]	740	DLS	Smears of HIV-positive women	For the detection of cervical cellular atypia, sensitivities were 95.7% compared with the pathologist’s assessment of digital slides, and 100% compared with the pathologist’s assessment of physical slides. Specificities were 84.7% compared with the pathologist’s assessment of digital slides, and 78.4% compared with the pathologist assessment of physical slides. The corresponding AUCs were 0.94 and 0.96. Accuracy and NPV were both high, especially for the detection of high-grade lesions. Compared to the pathologist’s evaluation of digital slides, there was a significant level of interrater agreement.
Tan et al., 2021 [50]	13,775	Robust DCNN model	ThinPrep cytology test	With an AUC of 0.67, the proposed cervical cancer screening system had a sensitivity and specificity of 99.4% and 34.8%, respectively.
Tang et al., 2021 [51]	10,601 cases	Manual reading compared with AI assistance	Abnormal cervical epithelial cells	Sensitivity for the detection of LSIL and HSIL increased remarkably from 0.837 to 0.923 and 0.830 to 0.917, respectively.
Wang et al., 2021 [52]	143 images	DL-based cervical lesions diagnosis system	De-identified, digitized whole-slide images of conventional Pap smear samples	A high precision (0.93), recall (0.90), F-measure (0.88), and Jaccard index (0.84) were achieved with the DL-based technique. According to the run-time analysis, the suggested technique processes a WSI in only 210 s, which is 20 times faster than U-Net and 19 times faster than SegNet.
Wentzensen et al., 2021 [53]	4253 patients	Cloud-based whole-slide imaging platform with a deep-learning classifier compared with conventional Pap and manual DS	Cervical images	AI-based DS had a lower positive rate than cytology and manual DS, equal sensitivity, and much higher specificity when compared to both Pap and manual DS. When compared to Pap, AI-based DS reduced referrals to colposcopy by 31% (41.9% vs. 60.1%).
Fu et al., 2022 [54]	2160 women	DL	Cervical images	With an AUC of 0.921, the cross-modal integrated model achieved the best performance.

Abbreviations: AI: artificial intelligence, ASCUS: atypical squamous cells of undetermined significance, AUC-ROC: area under the curve—a receiver operating characteristic curve, AVE: automated visual evaluation, CIN: cervical intra-epithelial neoplasia, CNN: convolutional neural network, DCNN: deep convolutional neural network, DL: deep learning, DLS: deep learning system, DS: dual-stained, DT: decision tree, HSIL: high grade squamous intra-epithelial lesion, LSIL: low-grade squamous intraepithelial lesion, KNN: k-nearest neighbor, LD: linear discriminant, Mask-RCNN: mask region based convolutional neural network, MFDFA: multifractal detrended fluctuation analysis, NNA: neural network accelerator, NPV: negative predictive values, PPV: positive predictive values, RF: random Forest, ROC: receiver operating characteristic curve, SIL: squamous intraepithelial lesions, SVM: support vector machine, VIA: visual inspection of the cervix with acetic acid, WSI: network-based whole slide image, YOLOv3: You Only Look Once.

**Table 3 diagnostics-12-02771-t003:** Characteristics of studies on the use AI in cytology to detect cervical cancer.

First Author/Year	Sample Size	Methods	Datasets	Main Results
Sherman et al., 1994 [101]	20	PAPNET system vs. conventional microscopy screening	Cervical smears	Each PAPNET analysis (conducted by pathologists) identified SILs in 10 individuals who were missed in the initial screening and selected smears for rescreening in 19 (95%) of 20 patients.
Kok and Boon, 1996 [102]	25,767 conventional and 65,527 PAPNET smears	PAPNET system vs. conventional screening	Cervical smears	The consistency of screening was much higher for PAPNET than for traditional screening with regard to invasive cancer and high-grade SIL smears. A higher screening sensitivity was demonstrated by the higher positive results for invasive and in situ cancer on histology.
Cenci et al., 1997 [103]	3000	PAPNET system	Conventional cervical smears	PAPNET detects false negative cytological errors rapidly and accurately.
Doornewaard et al., 1997 [64]	46 cases, 920 control smears	PAPNET system	Histologically confirmed CIN3 or carcinoma	Twenty percent of negative smears were positive. Two were reclassified as high-grade and seven as low-grade squamous intraepithelial lesions. In the 920 smears that constituted the control group, 1 of the 31 initially positive smears was misidentified. Fourteen newly discovered positive cases (1.6%) were found in the control group of 889 negative smears; all of these were low-grade SIL.
Kemp et al., 1997 [104]	344 slides for cell-by-cell classification, 395 slides for slide-by-slide classification	Linear discriminant functions, feed-forward neural networks, Quickprop algorithm	Conventional spatula-collected cervical cell smears	For the test data, a linear discriminant function had an accurate classification rate of 61.6%, whereas neural networks had a cell-by-cell score of up to 72.5%. Neural networks achieved a high rate of 76.2% valid classifications, and the discriminant function achieved a mere 67.6%.
Koss et al., 1997 [105]	487 negative smears	PAPNET testing system	Archival negative smears (index smears) from 228 women with biopsy-proven high-grade precancerous lesions or invasive cervical carcinoma	PAPNET enhanced the detection rate of SILs in control smears by 25% and raised the yield of quality control rescreening by 5.1 times when compared to historical performance data from various participating laboratories.
Keenan et al., 2000 [106]	230	ML	Smears	62.3% of the CIN cases had the proper category assigned to them.
Dickman et al., 2001 [107]	8 training images, 8 test images	GTV system	Cervigrams, 35-mm colpophotographs and direct computer-captured colposcopy images	GTV achieved 100% sensitivity and 98% specificity in detecting CIN3 after being trained on one set of photos and tested on another set of images. Following training on one set of digitized cervical colposcopy pictures and testing on another set of images, GTV also achieved a sensitivity of 88% and a specificity of 93% for the detection of cervical cancer.
Giovagnoli et al., 2002 [65]	12 FNs	PAPNET	Cervical smears	When used in cervical screening, Nnbt can assist the diagnosis of misread smears in addition to allowing the detection of FNs due to screening errors.
Parker et al., 2002 [108]	17 women	Neural net	Abnormal Papanicolaou smears	Average correct classification rates for the intrapatient and interpatient nets were 96.5% and 97.5%, respectively. For grade I cervical intraepithelial neoplasia, the sensitivity, specificity, positive predictive value, and negative predictive value were 98.2%, 98.9%, 71.4%, and 99.9%, respectively.
Boon et al., 2005 [109]	1010	Neural network scanner: PAPNET	Cervical cell samples suspended in the coagulant fixative BoonFix in liquid-based PapSpin slides	A change in the diagnostic parameter was noticed on the PapSpin slide for 151 of 151 exceptional cases, or 85%. In 94% of the cases, it was simpler to determine whether inflammatory cells were adherent to epithelial cells, whereas the adhesion of microbes varied between 43% and 100%.
Dounias et al., 2006 [110]	500	Hard C-means/fuzzy C-means/Gustafson–Kessel clustering/feature selection/ANFIS neuro-fuzzy classification/nearest neighbor classification/entropy information-based inductive machine learning/genetic programming-derived crisp rule-based system/(LMAM/OLMAM) type second order neural networks	Pap-smear images collected automatically with the aid of software especially designed to recognize, under the electronic microscope, the regions of nucleus-cytoplasm-background.	In the 2-class problem, the vast majority of the techniques performed exceptionally well, frequently achieving a test accuracy of 90%. However, in the 7-class problem, most of the techniques only achieved an average testing accuracy of approximately 75%. Genetic programming demonstrated the best average generalization capabilities in both types of issues considered, achieving 89% and 81% accuracy for the 2- and 7-class problems, respectively. Second-order neural networks scored highest in the 2-class problem, with an accuracy of 99%.
Mat-Isa et al., 2008 [111]	550	A new artificial neural network architecture known as hierarchical hybrid multilayered perceptron	Pap smears	The proposed network achieved 96.67% sensitivity, 100% specificity, and 97.50% accuracy. False positives and negatives were 1.33% and 3.00%, respectively.
Wang et al., 2009 [112]	31 available digital slides	SVMs	Cytology images	Initial findings point to the system’s potential as a training and diagnostic tool for pathologists.
Al-Batah et al., 2014 [113]	500	Moving 𝑘-mean, SBRG, ANFIS	Single cell images captured from the slides by using the AutoCapture system	Based on the five-fold analysis method, MANFIS produced a training accuracy rate of 96.3% and a testing accuracy rate of 94.2%.
Sokouti et al., 2014 [114]	100 patients	LMFFNN	Cervical cell images	Using the suggested strategy, cervical cell images were successfully classified at a 100% correct classification rate. Additionally, using the LMFFNN technique, the rates of sensitivity and specificity were 100%. Good concurrence was noted between the values obtained from the ANN model and the expert decision.
Kim et al., 2015 [115]	30	Image processing by the Hough transform extraction algorithm	Cell images	Using a liquid-based cytology software, the accuracy was 91.5%. The software’s Hough transform extraction algorithm evaluation yielded a success rate of 95%. The Hough transform extraction technique was found to have potential advantages over extraction algorithms for imaging.
Kyrgiou, et al., 2016 [116]	3561 patients	ANN implemented by a MLP	Detailed patient characteristics and the colposcopic impression.	The sensitivity for predicting CIN2 or worse was 93.0%, the specificity was 99.2%, and the positive and negative predictive values were also high (93.3% and 99.2%, respectively).
Hyeon et al., 2017 [117]	71,344	CNN as feature extractor/classifiers: LR, RF, AadaBoost, SVM	Pap smear microscopic images from Seegene Medical Foundation	SVM performed the best, achieving an F1 score of 78%.
Abdoh et al., 2018 [118]	858	RF, feature reduction, recursive feature elimination, PCA	Historical medical records, habits, and demographic information	With regard to all features, the SMOTE-RF model had the best accuracy, sensitivity, PPA, and NPA. The SMOTE method is able to increase sensitivity and PPA ratios. For all target variables, sensitivity increased from 86% to 96% and PPA increased from 30% to 98%.
Arya et al., 2018 [119]	330 and 917	Texture-based feature extraction/classifiers: ANN, SVM	Generated dataset MNITJ (330), DTU/Herlev Pap smear benchmark dataset (917)	With the help of ANN, the suggested texture features technique achieved 99.50% accuracy, 99.90% sensitivity, and 99.90% specificity. For the categorization of single cell images, an accuracy of 99% was achieved using the SVM quadratic classifier, with a sensitivity and specificity of 98.04% and 98.00%, respectively.
Aljakouch et al., 2019 [120]	-	DCCN	Pap-smears	The distinction between healthy and malignant Pap smears was made with 100% accuracy by DCNNs based on CARS, SHG/TPF, or Raman images.
Bhuvaneshwari and Poornima, 2019 [121]	20 Pap smear images	Fuzzy c means segmentation, k- k-NN classifier	The single cell microscopic image data were collected from cancer registry hospitals.	On multi-cell and overlapped cells, the approach works quite well. For the KNN classifier this technique achieved a precision of 95%.
Lasyk, et al., 2019 [122]	2058	U-NET and CNN	Liquid-based cytology samples	Normal and abnormal samples could be distinguished with 100% sensitivity and specificity.
Ma et al., 2019 [123]	92 patients, 141, 467 images	CNN and SVM	Gray-scale cervical tissue images	The classification accuracy for five groups of cervical tissue-normal, ectropion, LSIL and HSIL, and cancer-was 88.3%. The approach yielded an area-under-the-curve value of 0.959 in the binary classification [low-risk (normal, ectropion, and LSIL) against HSIL and cancer] with a sensitivity and specificity of 86.7% and 93.5%, respectively.
Moscon et al., 2019 [124]	15	Machine-based learning image	Samples of cervix cells	A high sensitivity (99%, 99%) and specificity (98%, 97%) was noted for distinguishing normal cells and HSIL. However, sensitivity (78%) and specificity (79%) were lower for LSIL cells.
Wang et al., 2019 [125]	917	Deep network model	Cervical cytology images	The experimental results demonstrated that the lightweight deep model performs better than the previous compared models and is able to obtain a model accuracy of 94.1% when applied to a cervical cell dataset with fewer parameters.
Zhang et al., 2019 [126]	62	R-FCN	Cervical cell images	According to experimental findings, detecting abnormal regions in cervical smear images is accomplished with an average precision of 93.2%. The suggested approach shows promise for the creation of computer-aided cervical cytological screening systems.
Bao et al., 2020 [66]	188,542	CNN	Digital cytological images from database of routine screening	Compared to manual reading, AI-assisted reading recognized 92.6% of CIN 2 and 96.1% of CIN 3+. AI-assisted reading showed higher specificity (relative specificity 1.36) and equal sensitivity (relative specificity 1.01) compared to expert cytologists, but higher specificity (1.12) and sensitivity (1.12) compared to cytology doctors.
Guruvare et al., 2020 [127]	66	PNN classifier, the exhaustive search feature selection method, the leave-one-out and the bootstrap validation methods	Microscopy images of H&E-stained biopsy material from two different medical centers	The accuracy of the pattern recognition system was 93% and 88.6% when using the leave-one-out and bootstrap validation methods, respectively.
Ma et al., 2020 [128]	4107	Cervical cancer detection booster based on FPN and Retinanet	Slide images of cervical smears	The sensitivity of the suggested technique at four false positives per image and the average precision were both increased compared to baseline (Retinanet) by 2.79% and 7.2%, respectively.
Xia et al., 2020 [129]	4036	SPFNet	Cervical cytology images	The experimental findings demonstrated that the framework outperformed more traditional detection methods by 78.4% AP in cervical cancer cell identification tests.
Ali et al., 2021 [80]	-	RF, IBK/feature selection techniques	Kaggle data repository for cervical cancer	The best results were achieved by RF and IBk for Hinselmann (99.16%) and Schiller (98.58%), respectively.
Cao et al., 2021 [130]	325	CNN, vs. Faster R-CNN	ThinPrep Pap test slide datasets	An independent testing dataset with 3970 cervical cytology images achieved an overall sensitivity, specificity, accuracy, and AUC of 95.83%, 94.81%, 95.08%, and 0.991, respectively, which is comparable to a pathologist with 10 years of expertise. The feature pyramid network model is almost 380 times faster than an average pathologist.
Diniz et al., 2021 [75]	45 training imagesand 900 test images	DT, Nearest Centroid, and k-NN	Cervical cytology images	The suggested ensemble method maintained high precision while achieving the highest results in terms of F1 (0.993) and recall values (0.999).
Jia et al., 2021 [131]	1462	SSD	Benchmarked cervical cells dataset	The accuracy and mAP of the suggested optimized SSD network were 90.8% and 81.53%, respectively, which is 7.54% and 4.92% higher than YOLO and conventional SSD, respectively.
Li et al., 2021 [132]	800	Novel framework based on Faster RCNN-FPN	Cervical image dataset	With a mAP of 0.505 and an AUC of 0.670, the proposed model is superior to all other state-of-the-art models. When integrated with traditional computer vision approaches for tagging the negative picture samples, the mAP increased by 6–9%.
Liang. et al., 2021 [133]	12,909 cervical images with 58,995 ground truth boxes corresponding to 10 categories objects	A global context-aware network based on YOLOv3 algorithm	Cervical cell dataset	With the sacrifice of a 2.6% delay in inference time, the suggested approaches ultimately achieve increases a mAP of 5.7% and specificity of 18.5%.
Liang et al., 2021 [134]	7410 and a small-sized dataset of 762 randomly selected images	Faster R-CNN with FPN	Cervical microscopic images	With a mAP of 26.3%, the suggested comparison detector improved on the small dataset. Using the medium-sized dataset for training, the comparison detector improved its mAP by 48.8%.
Lin et al., 2021 [67]	19,303	CNN with dual-path encoder	Cervical slide images from multiple medical centers	The technique performed effectively, with a high sensitivity of 0.907 and a specificity of 0.80.
Meng et al., 2021 [74]	100 slides from 71 patients	MobileNet-v2, VGG, GoogLeNet, Inception-v3, DenseNet, and ResNet/segmentation networks including FCN, SegNet, DeepLab v3+, U-Net, HookNet	Cervical histopathology image dataset	The dice coefficient approaches 0.7833, showing that the suggested weakly supervised ensemble technique is effective.
Pal et al., 2021 [135]	1331 images	Multiple instance learning	Cervical histopathology images	A framework for multiple instance learning with sparse attention that can provide a classification accuracy of up to 84.55% on the test set.
Sheela Shiney et al., 2021 [68]	-	AMBSS algorithm and SVM	Pap images	The achieved accuracy was 85.4%. AMBSS with quasi-Newton-based feedforward neural network classification was employed to increase accuracy, and a classification accuracy of 96.0% was achieved. Additionally, The AMBSS classification using a deep auto encoder-based extreme learning machine achieved an accuracy rate of 99.1%.
Jia et al., 2022 [136]		YOLO algorithm, improved algorithm k-means++ is used to replace the clustering algorithm k-means in the original yolov3, NMS algorithm		Experimental verification showed that the network achieved a mAP of 78.87% which is 8.02%, 8.22%, and 4.83% higher than that of SSD, YOLOv3, and ResNet50, respectively.

Abbreviations: AMBSS: advance map-based superpixel segmentation, ANN: artificial neural network, ANFIS: adaptive neuro-fuzzy inference system, ASCUS: atypical squamous cells of undetermined significance, CIN: cervical intraepithelial neoplasia, CNN: convolutional neural network, DCNN: deep convolutional neural networks, DT: decision tree, FCN: convolutional network, FPN: feature pyramid network, FNs: false negatives, GTV: Georgia tech vision, HSIL: high-grade squamous intraepithelial lesion, IBK: instance-based learning with parameter k, KNN: k-nearest neighbor, LSIL: low-grade squamous intraepithelial lesion, LMAM: Levenberg–Marquardt with adaptive momentum, LR: LMFFNN: Levenberg–Marquardt feedforward MLP neural network, H&E: hematoxylin and eosin, MANFIS: multiple adaptive neuro-fuzzy inference system, ML: machine learning, mAP: mean average precision, MLP: multilayer perceptron, NC: normal cell, NPA: negative predicted accuracy, nnbt: neural network–based technology. NMS: non-maximum suppression, OLMAM: optimized Levenberg–Marquardt with adaptive momentum, PCA: principal component analysis, PNN: probabilistic neural network, PPA: positive predicted accuracy, RF: random forest, R-CNN: region-based convolutional neural networks, RFCN: region-based fully convolutional network, SIL: squamous intraepithelial lesion, SSD: single-shot multibox detector, SMOTE: synthetic minority oversampling technique, SPFNet: series-parallel fusion network, SVMs: support vector machines, SBRG: seed-based region growing, YOLOv3: You Only Look Once.

**Table 4 diagnostics-12-02771-t004:** Characteristics of studies on the use of AI in colposcopy for the detection of cervical cancer.

First Author/Year	Sample Size	Methods	Datasets	Main Results
Park et al., 2008 [139]	29 patients	K-means clustering algorithm	Digital images of the cervix	Diagnostic performance: 88% specificity and 79% sensitivity.
Li et al., 2009 [140]	99 human subjects	Automated image analysis	Images captured with a digital colposcope	The proposed opacity index demonstrated 94% and 87% sensitivity and specificity, respectively.
Park et al., 2011 [141]	48 patients	CRFs	Clinical data	The suggested automated diagnostic approach can supplement or even replace conventional colposcopy, permit more objective tissue specimen sampling, and reduce the incidence of cervical cancer in low-income nations by offering an economical screening option.
Ramapraba et al., 2017 [143]	400 images	DWT and KNN	Cervical images	The cervical acetowhite lesion can be found with 94% sensitivity in less than 40 s.
Asiedu et al., 2019 [137]	134	ML	Pocket colposcope patients	The suggested framework successfully distinguished cervical intraepithelial neoplasia (CIN+) from benign and normal tissue with sensitivity, specificity, and accuracy rates of 81.3%, 78.6%, and 80.0%, respectively. This is better than the average values obtained by three expert doctors on the same dataset (77% sensitivity, 51% specificity, and 63% accuracy) for differentiating normal/benign cases from CIN+.
Bai et al., 2020 [69]	817	CLDNet	6536 Colposcopy images from attendees of cancer screening	The average precision of the model extraction lesion region is 92.53%, and the average recall rate is 85.56%.
Cho et al., 2020 [144]	6000 cases	Pre-trained CNN	Photographs, colposcopy-directed biopsy and conization	The AUC of the CIN system for differentiating high-risk lesions from low-risk lesions by Resnet-152 was 0.781 and the AUC of the LAST system was 0.708.
Li et al., 2020 [145]	7668	GCNs	Colposcopy images	Similar to an in-service colposcopist, the suggested deep learning framework achieves a classification accuracy of 78.33%.
Luo et al., 2020 [146]	Positive samples of 494 cases, negative samples of 615 cases	Multi-CNN	Colposcopy images obtained through a lighted magnifying glass and clinical diagnosis reports	Results with two data splits were compared as follows: single-class split: AUC = 0.756; multi-class split: AUC = 0.764. The suggested multi-decision feature fusion technique can produce computer-aided diagnosis outcomes that are more in line with clinical diagnosis requirements.
Miyagi et al., 2020 [70]	330 patients	DL	Colposcopy images	For diagnosing HSI, the AI classifier and oncologists performed with accuracy, sensitivity, specificity rates, and a Youden’s J index of 0.823 and 0.797, 0.800 and 0.831, 0.882 and 0.773, and 0.682 and 0.604, respectively. The 95% confidence interval for the area under the receiver-operating characteristic curve was 0.721–0.928. The best cutoff value was 0.692.
Xue et al., 2020 [147]	19,435 patients	CAIADS	Colposcopy images, clinical information, and pathological results	While the specificities were comparable (low-grade or worse 51.8% vs. 52.0%; high-grade or worse 93.9% vs. 94.9%), CAIADS demonstrated higher sensitivity for detecting pathological HSIL+ than colposcopies interpreted by colposcopists at either biopsy threshold (low-grade or worse 90.5% vs. 83.5%; high-grade or worse 71.9% vs. 60.4%).
Yuan et al., 2020 [148]	5384	ResNet	Colposcopy images of three subsets: the training set, the test set and the validation set, in a ratio of 8:1:1	With an AUC of 0.93, the classification model’s sensitivity, specificity, and accuracy in differentiating between negative and positive cases were 85.38%, 82.62%, and 84.10%, respectively.
Yue et al., 2020 [149]	4753	C-RCNN	Cervigram images	Achieved a test accuracy of 96.13%, specificity of 98.22%, and sensitivity of 95.09%. The AUC was more than 0.94.
Adweb et al., 2021 [150]	4000 pre-cancerous, 800 healthy samples	ReLU-ResNet, PReLU-ResNet and Leaky-ReLU	Cervical images from colposcopy	The accuracy of designed residual networks with leaky and parametric rectified linear unit (Leaky-RELU and PRELU) activation functions (accuracies of 90.2% and 100%, respectively) was similar.
Chandran et al., 2021 [151]	5679	CYENET and VGG19 (TL)	Colposcopy photographs	For VGG19, the classification accuracy was 73.3%. High sensitivity, specificity, and kappa scores of 92.4%, 96.2%, and 88% were achieved by the proposed CYENET.
Hunt et al., 2021 [152]	1486	Multi-task CNN	High-resolution microendoscopy images	For the detection of CIN3+, HRME with morphologic image analysis was just as sensitive (95.6% vs. 96.2%) and specific (56.6% vs. 58.7%) as colposcopy. Compared to colposcopy, HRME with morphologic image analysis had a slightly lower sensitivity (91.7% vs. 95.6%) and specificity (59.7% vs. 63.4%, *p* = 0.02) for the identification of CIN2+.
Li et al., 2021 [153]	8604	CAD systems	Colposcopy images from grading cervical intraepithelial neoplasia	The grading accuracy of CIN increased by more than 10% with TAE and ABV.
Nikookar et al., 2021 [71]	287	NavieBayes, AdaBoost, RF, R tree, SVM, Decision tree, Logit boost	Cervigrams from digital colposcopy dataset	Random tree is the best performing classifier on the dataset acquired by applying the F_mvs_ aggregation function.
Peng et al., 2021 [72]	960	CNN	Original colposcopy image	In 60 tests, the suggested technique achieved a classification accuracy of 86.3%, sensitivity of 84.1%, and specificity of 89.8%.
Viñals et al., 2021 [73]	21,851 positive pixels and 93,725 negative pixels	ANN	VIA videos	The sensitivity and specificity of the suggested solution were 0.9 and 0.87, respectively.
Yan et al., 2021 [154]	1400	BF-CNN vs. ResNet18	Cervicograms	Similar sensitivity (74.6%) and the best accuracy (85.5%), specificity (95.7%), and AUC (0.909) were achieved with F-CNN.
Yue et al., 2021 [155]	609	CICN	Clinical cervigram	DenseNet-121 achieved the highest accuracy (0.906) and AUC (0.973)
Kim et al., 2022 [79]	234 patients	ML	Cervical images	Compared to each clinician’s colposcopic impressions, AI was associated with greater sensitivity, equivalent specificity, and equivalent positive predictive value.
Elakkiya et al., 2022 [77]	858	SOD-GAN	Cervical samples and colposcopy images	The suggested method demonstrated good accuracy through all stages, achieving a sensitivity of approximately 97% with a loss of less than 1%.
Ito et al., 2022 [156]	463	AISD	Colposcopy images	The accuracy of AI was 57.8% for normal, 35.4% for cervical intraepithelial neoplasia (CIN)1, 40.5% for CIN2–3, and 44.2% for invasive cancer. Before learning about the AI image diagnosis, the accuracy of gynecologists’ diagnoses based on cervical pathology images was 54.4% for CIN2–3 and 38.9% for invasive cancer. Their accuracy increased to 58.0% for CIN2–3 and 48.5% for invasive cancer after they learned about the AISD.
Zimmer-Stelmach et al., 2022 [138]	48	AI colposcopy assessment	Colposcopy examinations	With a significantly lower sensitivity (66.7% vs. 100%) but a higher specificity (46.7% vs. 16.7%), AI-assisted colposcopy was able to detect diseases with a similar PPV as that of a skilled physician (42.9% vs. 41.8%).

Abbreviations: AISD: AI-image-assisted diagnosis, AUC: area under the receiver operating characteristic curve, BF-CNN: bilinear fuse convolutional neural network, CAD: computer aided diagnosis, CAIADS: colposcopy artificial intelligence auxiliary diagnostic system, CICN: computational intelligence and communication networks, CIN: cervical intraepithelial neoplasia, CLDNet: cervical lesion detection net, CNN: convolutional neural network, C-RCNN: cervigram-based recurrent convolutional neural network, CRFs: conditional random fields, DL: deep learning, DWT: discrete wavelet transform: GCNs: graph convolutional network, HSIL, high-grade squamous intraepithelial lesion, KNN: k-nearest neighbor, ML: machine learning, ResNet: residual neural network, Rtree, random tree, RF: random forest, SOD-GAN: small-object detection generative adversarial networks, VIA: visual inspection with acetic acid.

## Data Availability

The datasets analyzed for the current study are available from the corresponding author on reasonable request.

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
