# Peer review of "Diagnosis of Cervical Cancer and Pre-Cancerous Lesions by Artificial Intelligence: A Systematic Review"

_diagnostics, 2022, doi:10.3390/diagnostics12112771_

Round 1

Reviewer 1 Report

See below for specific comments;

-Greater depth on the epidemiological of cervical cancer is required

-The tables are good and relevant, but much more discussions and synthesis is required

-Oppurtunities for further work in the area needs to be churned out from the review and included towards the conclusion

-Discussions and Refs should be expanded to synthesise clinical decision support work done by the likes of Nsugbe et al, as well as Topol et al

Author Response

Thank you for this valuable comment. Introduction section was revised.

Discussion was revised.

Please refer discussion section.

Opportunities for further work in…. sentences came in discussion section were moved to conclusion section.  

Thank you for mentioning those valuable studies,

We strengthened the discussion section by using of those studies.

 Topol EJ. High-performance medicine: the convergence of human and artificial intelligence. Nat Med. 2019;25(1):44-56.

Nsugbe E. Towards the use of cybernetics for an enhanced cervical cancer care strategy. Intelligent Medicine. 2022;2(3):117-26.

Reviewer 2 Report

The manuscript represents a survey. Its main aim is to find responses for the following question : what is the accuracy of AI techniques in detecting cervical cancer or pre-cancerous lesions?

Studies on several datasets were investigated and compared.

The strategy of selection of the studied papers is explained in the manuscript (abstract, section 2, Fig. 1). 

However, here are some recommendations to enhance the quality of the study:

1. Table 1 : I suggest to add a column to illustate the drawbacks for each study.

2- What is the difference between Table 1 and Table 2. Please highlight the difference between "predicting" and "screening" cervical cancer.

 3. Table 1, 2, 3 : Please add more studies form 2022 (in each table, only one study from dozen is published in 2022)

4. Table 3 : Table 3 contains 14 studies published between 1994 and 2009. Please use more recent studies.

5. Line 282: Authors said "In the present report we reviewed ..." what report?

 6. Experimental tests on real datasets wouÙ…d better highlight the differences betweØ«n the investigated studies and validate the findings discussed in section 4.

7. It is highly recommended to indicate the interest of applying intelligent AI methods for the analysis of task scheduling problem. Here some studies to refer to, highlighting the interest of AI methods for complex real-world problems such yours: https://doi.org/10.1109/IACS.2019.8809127 and https://doi.org/10.3390/info9110284

8. Section 4 should give and explain the main open issues and perspectives of the studied problem.

Author Response

Thank you for your valuable comments. We did our best to consider and apply all your comments which strengthened the quality of this manuscript.

Thanks for this comment. We added the drawback column, unfortunately most studies did not mention to their studies’ drawbacks. Please refer to Table 1.

Thank you for this comment.

We explained the difference between Table 1 and Table 2 in the text below the titer 3.1.2 and 3.1.3. (Pages 5, lines 212-215), (pages 9 lines 220-236).

Predicting:

Identifying predictors is essential for making precise and meaningful prediction (1) Multivariate adjustment and multiple-regression techniques were introduced for prediction (that is, for estimating the predicted value of a certain outcome as a function of given values of independent variables) (1).

 Screening cervical cancer:

Screening is a way of identifying apparently healthy people who may have an increased risk of a particular condition (2). The screening test needs to be sensitive and precise. A screening test must have sensitivity exceeding 95% if the specificity is less than or equal to 95% and vice versa (specificity must be >95% if the sensitivity is 95%) in order to detect more true-positive cases than false-positive cases when the prevalence of the disease is less than or equal to 5% (which covers the majority of screening populations). Most screening tests do not meet this high standard, which means that the screening program must absorb the costs of many false-positive results (3).

The research team has re-searched, focusing on studies published during 2021-2022, unfortunately, no new studies were found which met the inclusion criteria to be included in present review study.

Thank you for this valuable comment.

Since comprehensive searches were performed on databases to find papers published until July 2022. Articles that applied any AI technique for the prediction, screening and diagnosis of cervical cancer and no time restriction was applied. So we included all articles which met the inclusion criteria in the study (from the 1994 until 2022).

We updated our search in revision phase with focusing on recent years, unfortunately, no new studies were found which meet the inclusion criteria to be included in the study.

Thank you for the precise comment. It was a typo error and it was revised.

In recommendation section of this review, we applied this comment.

Thanks, sentences from those valuable studies were added.

Thanks for this great point. We reviewed the problems of studies and explained limitation of studies.

Reviewer 3 Report

The topic of the review is interesting. However, it needs some changes to be incorporated for further enhancement of the presentation of the review.

1. The Introduction section needs to be further strengthened. Recent studies should be discussed that support the role of AI Technologies and their influence on the Diagnosis of Cervical Cancer and Pre-cancerous Lesions

2. Few recent studies are not included in the review like:

https://dx.doi.org/10.30919/es8d633

https://doi.org/10.1007/s12011-022-03226-2

https://doi.org/10.1158/1078-0432.CCR-21-3710

 https://doi.org/10.1002/ijgo.14486

3. Authors are suggested to recheck the screening process included in the selection of the articles for the review. The fonts use in the flowchart of PRISMA can be changed for better clarity.

4. The tables used for detailing the previous studies can be further elaborated with more columns included like Type of dataset, Information on Training, test and validation dataset used, Algorithms used, Outcomes of the study.

5. The conclusion section can be comprehensively rewritten to provide an overall conclusion of the review and its outcomes with critical observations on the best method observed from the literature and further directions provided to the researchers for future research.

Author Response

Thank you for this valuable comment. Introduction section was revised based on this comment.

Thank you for introducing those valuable studies.

Since the studies mentioned by the esteemed reviewer are review articles and one of inclusion criteria in this study was "non-review studies", we were not able to include those study in the present review.  But we used these studies to strengthen the introduction and discussion sections. 

Screening process in the selection of the articles were rechecked and revised. Font and style of flowchart diagram was revised.

Yes, thank you for this valuable comment. We had also decided to extract more data from the studies and listed them in tables. Unfortunately, the style of data reporting in the articles was very broad and varied.

We were able to extract the data including” “First Author/Year, Sample size, Methods, Datasets, Main results” which were common to all articles. We tried to list the studies drawbacks in Table 1(please refer to Table 1). As it turns out, most of the studies did not report any problems/drawbacks. But if the esteemed reviewer thinks it is okay for some data to be blank in the tables, limited cases reported in a limited number of studies can be added.

We revised the conclusion section.

We tried to consider reviewer’s concern in the both recommendation and conclusion sections.

Round 2

Reviewer 1 Report

Thanks for making the revisions